# A Sustainable Autoclaved Material Made of Glass Sand

**Anna Stepien** [1,*] , **Magdalena Leśniak** [2] **and Maciej SITARZ** [2]

1 Civil Engineering and Architecture Department, Kielce University of Technology, Al. 1000-lecia PP 7, 25-314 Kielce, Poland

2 Faculty of Materilas Science and Ceramics, AGH University of Science and Technology, Al. Adama Mickiewicza 30, 30-059 Kraków, Poland; mlesniak@agh.edu.pl (M.L.); msitarz@agh.edu.pl (M.S.)

* Correspondence: ana_stepien@wp.pl; Tel.: +48-41-34-24-479

**Abstract:** Far-reaching technological progress, manufacturing, and rapidly advancing globalization dictate new conditions for the development and changes in the construction industry. Valorization of by-products and the use of secondary materials in the production of building materials have attracted a lot of attention. Silicate materials were assessed on the basis of their compressive property. An orthogonal compositional plan type 3k (with k = 2), that is, a full two-factor experiment was applied in order to carry out the compressive strength and bulk density tests. Glass sand was added to the silicate mass as a modification. The results show that the compressive strength was higher than that of traditional bricks. Scanning electron microscopy coupled with energy dispersive spectrometry SEM/EDS was used to study the microstructure, whereas the XRD analysis was applied to examine the structures. Laboratory tests were performed on samples with dimensions of $50 \times 50 \times 50$ mm. The results show the bulk density increase to the value of 1.75 kg/dm$^{3}$, which increases the acoustic performance of the new products. The results of the modifications also indicate changes in the structure of the new bricks. The reference sample contained $\alpha$-quartz, zeolite, tobermorite 9A, and calcium aluminum silicate ($Ca_2Al_4Si_{12}O_{32}$), whereas the samples modified with glass sand, the presence of phases such as $\alpha$-cristobalite, natrolite, tobermorite 11A, gyrolite, and analcite was recorded.

**Keywords:** bricks; glass; lime; sand; quartz; amorphous materials; microstructure; crystallization

## 1. Introduction

Physical and mechanical properties of building materials depend on the composition, appropriate proportions, and quality of individual substrates involved in their production. Sustainability became a driving force behind the development of new building materials and products, including silicate bricks. Autoclaved sand–lime products are a special material because they are completely natural, neutral, and safe for the environment. These bricks are sustainable and offer many advantages with respect to the traditional "red" ceramic bricks (for example: high compressive strength within 15–25 MPa depending on the material class, high material density that promotes proper acoustics isolation in rooms, highest fire resistance class, lower water absorption (max. 16%).

The main objective of this paper is to present the properties of new types of sand–lime bricks made of glass components and to determine which basic properties occurred during and after the replacement of crystalline quartz sand (90% OS) by recycled glass sand (90% GS). Glass sand has an amorphous structure. The durability of these materials depends on the phase and thermodynamic transformations that occur in their structure (presence and number of crystalline and amorphous phases). Along with industrial and technological changes, the soil substrate geology changes in built-up areas. Therefore, there is a concern that the changed mineralogical composition may be unstable due to the nature of

the changes and will also affect the characteristics of concrete and other building materials [1]. The C–S–H phase, the basis for the structure of concrete materials, is thermodynamically stable in the given temperature range (mainly up to 25 °C). The hydration of Portland cement at or near ambient temperature produces more than 50% of a calcium silicate hydrate (C–S–H phase). According to Glasser's statement, the Ca/Si ratio of this nearly amorphous product can vary, ranging approximately from 0.8 to 2.0. Commercial Portland cement paste is usually characterized by the occurrence of crystalline $Ca(OH)_2$ together with higher C–S–H phase percentage and thus consists of a mixture of ordered and disordered constituents [2]. However, there is a group of materials subjected to high pressure already at the production stage which are sand–lime materials. Due to the low percentage of lime CaO, the main phase that builds the structure of this type of material is tobermorite [3]. Numerous additives introduced into the silicate mass disturb the internal structure, giving the possibility of crystallization to other phases (xonotlite, gyrolite, natrolite) [4–8]. Autoclaved sand–lime bricks are very popular in Poland, Germany, Spain, Slovakia, and other countries in Europe. These types of materials, commonly called "silicates products", are construction materials providing a solid structure and comfortable interior microclimate [9,10]. Modifications of sand–lime bricks aim at optimizing the production process and replacing the quartz sand with another component. This principle applies to sustainable development and the ability to limit the use of natural substrates to produce an "artificial stone", that is mortar. In fact, the construction sector uses billions of tons of materials each year. Some modifications (e.g., products with barite and basalt aggregate) have improved the compressive strength up to around 41.3 MPa. Water absorption due to capillary action has been limited to 12%. Therefore, the use of by-products can be also an interesting alternative to prevent excessive environmental destruction (e.g., aggregates from demolition or reconstruction of buildings or plastic components such as polystyrene HIPS, polymers, etc.). The initial phase of their production involves mixing only substrates: CaO, $SiO_2$, and $H_2O$. The mixture is then placed in steel silos resembling reactors (the slaking lime process) [11–15]. The process of slaking is accompanied by an increase in temperature to around 60–80 °C. The standard lime/water ratio is 56 g to 18 g respectively [16]. At this stage, silica loses its crystalline structure, which in turn facilitates the subsequent formation of products. Then, the mixture is directed to the press, in which it is compressed at a pressure of 15–20 MPa, and formed into bricks and blocks. A hydraulic press is more effective for the production of this kind of bricks [17,18].

$$CaO + H_2O + SiO_2 => C\text{-}S\text{-}H \tag{1}$$

[The formation of C-S-H is followed by crystalline phases such as tobermorite (with low lime content), jennite (with a lot of lime), awfilite when temperatures rise]

$$Ca(OH)_2 + CO_2 \text{ (from air)} => CaCO_3 + H_2O \tag{2}$$

$$CO_2 + CaSiO_3 => CaCO_3 + SiO_2 \tag{3}$$

Calcium silicate hydrate $xCaO{\cdot}SiO_2{\cdot}yH_2O$ is a poorly crystalline thermodynamically metastable product of variable composition in terms of its $H_2O/SiO_2$ ratio and Ca/Si molar ratio [2,19]. In the final phase, the compressed blocks are placed in autoclaves and are subjected to a hardening process in the temperature of around 200 °C at the pressure of 1.6 MPa (232 PSI). During the next 6–12 h of autoclaving, the lime reacts chemically with sand and the mixture undergoes the process of recrystallization (usually it takes 8 h, 1 h heating, 8 h autoclaving, and 1-h cooling). The best lime for sand–lime bricks is the 'CL 90' lime. Sand–lime bricks are a chemical-free natural material (mortar). Glass is also a natural substrate. In the era of sustainable development, environmental benefits are sought. In this modification, the GS is used in the production of bricks because it is rich in lime and sodium (Na). Sodium can reduce the amount of lime during the production of bricks and natural quartz sand as well. Thus, the aim of this study is to check the applicability of glass in silicate products. The purpose of this modification is to limit the use of sand and lime in the silicate mass and reduce its production costs. By introducing the

appropriate mineral additives into the raw material, we cause external ions to appear in the reaction environment. This may either accelerate the transformation of the amorphous C–S–H phase into crystalline products such as tobermorite $Ca_5[3Si_3O_8(OH)]\cdot8H_2O$ (commonly autoclaved product) or xonotlite $Ca_6[3Si_6O_{17}](OH)_2$, or stabilize them, thus preventing further transformation into phases that adversely affect the strength of these product [15,20]. Glass components are applied mainly to concrete modification [21–28]. Analyses conducted by the SILICATY Group and others [29–32] prove that it is possible to limit the amount of lime up to 2% on the condition of good quality sand (rich in Al) or other modifiers are used. Very good results (in particular within the microstructure) are obtained in concrete when an amorphous component (fly ash, glass powder) is added. In silicate materials produced on an industrial scale, quartz, calcite, aragonite, wollastonite, and tobermorite are usually found. In the structure of the modified material, gyrolite (modeling of synthetic gyrolite) [33,34] and natrolite are the new synthesized phases. Simulations are carried out by changing the crystal lattice parameter for gyrolite or by changing structural parameters e.g., temperature displacement [34–37]. This kind of brick is produced in a process similar to that of autoclaved cellular concrete (ACC) production that differs from the traditional concrete production which requires hydrothermal conditions. In ACC, tobermorite is the crystalline phase but nearly amorphous gel calcium silicate hydrate (known as gel C–S–H, and next C–S–H phase) is also formed. Because of this, this study describes the relationship between the basic physio-mechanical and chemical properties of sand–lime materials which are typically subjected to hydrothermal treatment (autoclaving process) and which were modified through the introduction of glass components (glass sand "GS"). A summary of non-destructive methods for the glass industry and crystalline phases (e.g., tobermorite) were described by Bunaciu [38]. The use of glass components in concrete is worldwide, unlike the use of glass components in autoclaved bricks which needs further research. Aerated cellular concrete and calcium silicate hydrate phase are subject to special modifications and research (SEM, XRD) [5,39–41], while bricks are still a secondary material for this type of modification.

## 2. Methodology, Hydrothermal Conditions, and the Laboratory Tests

The samples of silicate bricks with dimensions of $50 \times 50 \times 50$ mm were prepared under the only laboratory conditions. The research was conducted on the basis of Polish standards [42]. Due to the possible operation of a laboratory autoclave, the process of autoclaving these laboratory bricks under the pressure was estimated for 5 h (plus 1 h to heat of the autoclave to a temperature of 200 °C and after 5 h of operation of the autoclave, the device was left to cool down). The autoclaving temperature was equal to 199–200 °C. Quartz sand (OS), glass sand (GS), lime (CaO) and water were used for the production of this type of bricks. The analysis involved the use also of the Statistica 10.0 program (compressive strength and bulk density especially for bricks with GS). A statistical approach was used in the experience and analysis of data. The multi-criterial techno-economic analysis was applied with 'multi-criterial exploration techniques'. To interpret the results, the method of analysis of the main components was introduced on the basis of the data, which contained a specification matrix of modifiers and used variables. When the established experimental plan was fulfilled, the data simulation was prepared [43]. Analytical programs can be used to simulate hydroterlamine processes [44]. Whereas the effect of the analyzed modifier on the tested product was defined based on SEM (Scanning Electron Microscopy-IROL 5400 with EDS spectrum and Hitachi S-3400) and X-ray diffraction analysis (XRD) measurements of powdered samples were conducted with Empyrean PANalytical diffractometer using Ka radiation from Cu anode. All measurements were performed with Bragg-Brentano setup at room temperature with the 0.0068 step size at 2 theta scanning range and the 145 s of measurement time for each step. Data analysis and the peak profile fitting procedure were carried out using Philips X'Pert HighScore Plus software. The elements were identified on the basis of the wavelength (X) or energy (E) of the Roentgen ray. The concentration of a particular element was determined by measuring its lines intensity. The SEM analysis was conducted in low and high vacuum. For the modifications of the bricks quartz industrial sand and glass sand (GS <80–160> microns) were used. On the basis of the images

obtained, the analysis of the microstructure and different phase composition of the tested products was possible. X-ray diffraction as a nondestructive technique for characterizing crystalline materials which give information on structures, phases, preferred crystal orientations, and other structural parameters, such as average grain size, crystallinity, strain, and crystal defects. For the SEM test, fragments up to 1 cm in size were obtained from $50 \times 50 \times 50$ mm bricks manufactured in laboratory conditions. The XRD test also collected fragments of material up to 1 cm in size from the same laboratory bricks, and then they were ground to powder form. Samples are derived from the same laboratory bricks. The area of the sample was swept by electron probe under voltage of 5–50 keV. On the basis of the images obtained, the analysis of the microstructure and different phase composition of the tested products was possible. The modifications of the bricks used were: quartz industrial sand and glass sand (GS). The phase analysis of the traditional brick and brick modified by sand glass samples were measured in the 5–70° range of 2. The research (SEM analysis) was conducted in low and high vacuum. XRF analysis is used to identify elements in a particular substance and to determine their amount (Table 1).

**Table 1.** Composition of the traditional sample and sample with glass sand.

| Ref. Sample | With GS |
| :---: | :---: |
| Si | Si |
| Ca | Ca |
| Mg | Na |
| O | Mg |
| C | Al |
| Al | O |
| Fe | C |

The concentration of a particular element is determined by measuring its lines intensity. XRF enables determining the elements of substances [45].

### 2.1. Sand–Lime Mixture

The traditional sand–lime mass was the basis for modification of products composed of sand (90%), slaked lime (7%), and water (3%). The sand used in the process has a grain size of 0–2 mm (90% relative to the weight of the product, wherein 50–60% of the 90% that sand with a grain size of 0–0.5 mm, and the remaining 30–40% of the 90% present in the sand mass is sand with a grain size 0.5–2 mm). For the modifications of the bricks ground glass from recycled bottles, glass sand (GS <80–160> microns, Figure 1) was used [46]. Components of GS used in the bricks are marked by higher density.

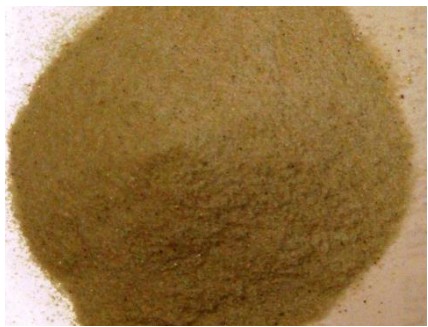

**Figure 1.** Glass sand (GS).

During the lime hydration process, in a mass modified with glass sand, a large number of "balls" are formed, which is difficult to mix (Figure 2). For this study, the temperature of lime hydration in the presence of crystalline sand was 81 °C (the temperature of reaction between quartz sand, lime (CaO), and water ($H_2O$)). However, the temperature of lime hydration in the presence of amorphous glass sand (GS) was 46 °C (above 35 °C degree difference between samples with traditional quartz sand (OS) and modification samples with recycled glass sand (GS)). The more glass sand was in the mass, the lime's hydration temperature was lower. Thirty test pieces were made with traditional quartz sand (Figure 3) and modified samples with different amounts of glass sand (Figure 4). Water in the amount of 7–9 % relative to the weight of the product is a supplement mixture (250 kg weight of the silicate is expected to approx. 18–20 L of water, giving 7.2–8% by weight relative to the weight of the product in industrial production). The examination was performed on forms of size 50 × 50 × 50 mm using the laboratory autoclave. Computer control was not subject to the slacking lime process, which affects chemical reactions and material durability.

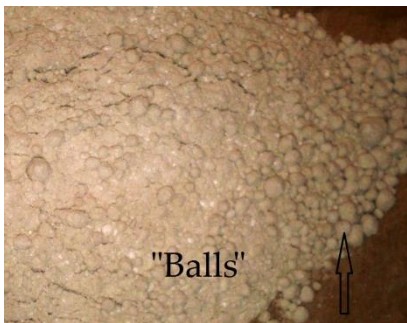

**Figure 2.** "Balls' in a mass modified by GS (CaO (7%) + GS (90%) + $H_2O$ (3%) and mixed and after the hydration of lime).

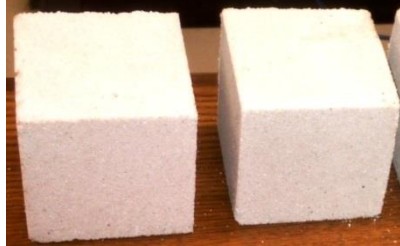

**Figure 3.** Sand–lime bricks with quartz sand (OS) after the autoclaving process.

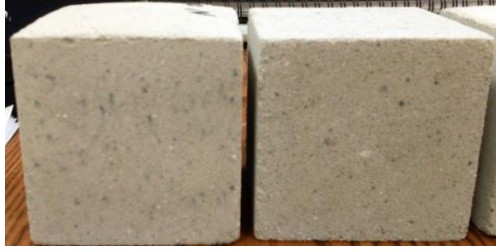

**Figure 4.** Sand–lime bricks with glass sand (GS) after the autoclaving process.

*2.2. Hydrothermal Conditions*

Traditional silicate products consist of sand, lime, and water and are chemically related to each other. The usual production process for silicate products is as follows. The mass (sand–lime and water) is mixed and then placed in steel silos tanks reactors. It is left in the reactor for around 2 to 4

h, as the process of slaking takes place, accompanied by an increase in temperature to around 60 °C. At this stage, silica loses its crystalline structure (the structure is weakened), which in turn facilitates the subsequent formation of products. In next step, the silicate mixture is directed to the press, in which it is compressed at a pressure of 15–20 MPa, and formed into blocks of suitable size and shape. Hydraulic presses are applied. In the final phase the compressed blocks are placed in autoclaves and subjected to a hardening process in the temperature of 200 °C at the pressure of 16 bar (1.6 MPa). The whole scheme was preserved (Figure 5), only the autoclaving time was changed to 5 h due to the autoclave operation (Figure 6).

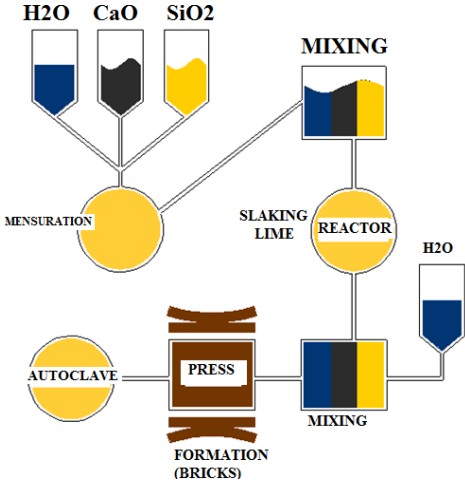

**Figure 5.** Simplified diagram of the production of the autoclaved material.

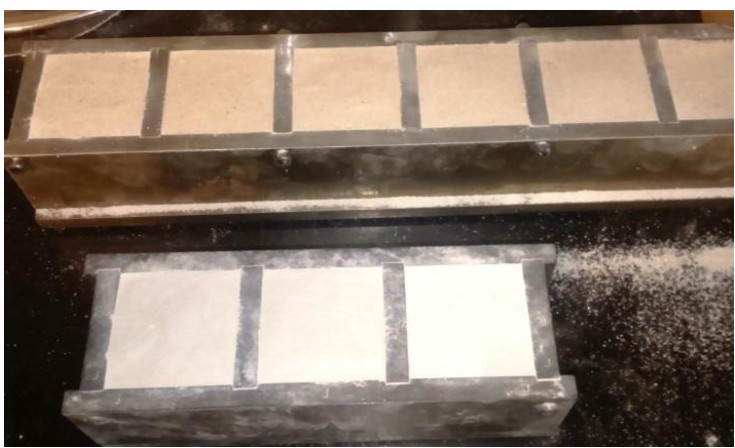

**Figure 6.** Sand–lime mass in forms before loading into the autoclave.

Figure 6 shows the molds with a fresh mixture (sand + lime + water) after lime hydration process and compression using a press, but before autoclaving process (according to Figure 5).

For analysis we used six representative samples from one series, which gave the most correct result and where the technological cycle was properly preserved, that is, the autoclave worked without problems, the hydration of lime reached a temperature of min. 42 °C for the modification of sand–lime mass by 90% GS during the hydration of lime.

### 2.3. Materials: Glass Sand Characteristics

The glass sand used in concrete and now in the production of bricks is nothing else than a ground glass cullet [46]. Addition in the form of glass sand is an amorphous metastable additive that changes its properties over time under the influence of pressure and temperature. Therefore, it is necessary to

thoroughly analyze the influence of glass sand not only on the physical and mechanical properties of the silicate brick but also on the durability of the material so modified. Components of glass sand GS used in the bricks are marked by higher density. Taking into account the benefits of the modification of glass brick, it can imply an improvement of acoustic isolation and density of the product. Mixed glass sand GS (glass sand 90- with a particle size of <80–160> microns, with sand $SiO_2$ (sand with a particle size of 0–0.5 mm 50–60% and sand 0.5–2 mm 40–50%) and water $H_2O$. During the lime extinguishing and mixing process, lumps (balls) are formed during the mass. The mixing process was not fully mechanized, which affected the production technology thus, the durability and physical, mechanical, and microstructural characteristics of these products. Thirty test pieces were made with different amounts of glass sand (Figure 4). The autoclaving time of the laboratory test was 5 h (due to the possibilities of a laboratory autoclave). From a chemical point of view, it is assumed that in the brick production process the following conditions should be met:

- the strength of sand–lime bricks depends on the temperature of the reaction between lime and sand ($CaO + SiO_2$), the quality of lime and sand, and the pressing process (compression);
- the amount of water depends on the moisture content of the sand;
- an activity CaO which is not less than 89.90%;
- $SiO_2$ sand containing at least 92% silica;
- the reaction temperature between CaO, $SiO_2$, and $H_2O$ should be 60–70 °C (and for quartz sand, this temperature is reached);
- compression of the fresh mass: 1.6–2.5 MPa;
- the temperature inside the autoclave (200 °C);
- the pressure inside the autoclave: 1.5–1.6 MPa.

Only laboratory tests were carried out and a correction to the conditions of production has to be taken into account. Computer control was not subject to the slacking lime process, which affects chemical reactions and material durability. The temperature of the reaction between glass sand (GS), lime (CaO) and water ($H_2O$) is max. 46° C (this is one of the differences between quartz sand used for production autoclaved bricks and glass sand we want to use).

In this paper, we present only a reference sample and a sample in which was completely eliminated quartz sand and replaced with glass sand (GS).

## 3. Results

On the basis of standards (PN-EN 772-13:2001) and the literature, as well as the known technological process, the compressive strength was established on the level of 15–20 MPa for the traditional sand–lime bricks. Bulk density of the autoclaved silicate products was established according to PN-EN 772-13:2001 standards [42]. The bulk density of traditional products is placed on the level of 1.7 kg/dm³. Traditional autoclaved silicate material absorbs water on the level of 16% water compared to its mass. The modified product should have no worse properties. The volume density (Equation (4)) and impregnability (Equation (5)) was tested according to the formula:

$$\rho_o = m_s/v_o \tag{4}$$

where:

$\rho_o$—bulk density,
$m_s$—the mass of the dry sample;
$v_o$—the volume of the sample

$$n_w = [(m_n - m)/m] * 100\% \tag{5}$$

where:

$n_w$—impregnability of weight;

$m_n$—the weight of the wet samples;

m—the weight of the dry samples.

### 3.1. Compressive Strength of the Bricks with GS

The experimental plan and the results of physical and mechanical tests were presented first. Analysis of the results was performed on the basis of statistical analysis (Figures 9 and 10). The tests for examining compressive strength [MPa] and density were carried out on the basis of a multi-criteria economic and engineering analysis, with a simultaneous determination of the optimum composition of the sand–lime mixture with the sustainable glass additions. The experiments are shown in Figures 9 and 10 and Table 2. The fractional plan (complete) 41 (at k = 1) is a full one-factorial experiment. The compressive simulation strength [MPa] was used as a response according to which the properties of silicate elements were assessed. An orthogonal compositional plan type 3k (with k = 2), that is, a full two-factor experiment was applied in order to carry out the experiments both in the compression strength test and bulk density test. For each factor correlation, six parallel tests were conducted. The methodology of experiment and obtained results are shown in Table 2. The compressive strength of the samples is described by regression equation:

$$6 = A_0 + A_1 (OS) + A_2 (OS)^2 + A_3 (GS) + A_4 (GS)^2 \tag{6}$$

**Table 2.** Plan of the experiment. Bricks with GS and traditional quartz sand (OS).

| Plan of the Experiment | | | | |
|---|---|---|---|---|
| **OS [%]** | **GS [%]** | **Compressive Strength [MPa]** | **Bulk Density [kg/dm$^3$]** | **Case** |
| 90 | 0 | 5.25 | 1.92 | OS 90% + G S0% |
| 80 | 10 | 5.11 | 1.97 | OS 80% + GS 10% |
| 70 | 20 | 14.25 | 2.09 | OS 70% + GS 20% |
| 60 | 30 | 15.03 | 2.00 | OS 60% + GS 30% |
| 50 | 40 | 14.13 | 2.18 | OS 50% + GS 40% |
| 40 | 50 | 18.31 | 2.21 | OS 40% + GS 50% |
| 30 | 60 | 17.47 | 2.01 | OS 30% + GS 60% |
| 20 | 70 | 19.50 | 2.25 | OS 20% + GS 70% |
| 0 | 90 | 20.23 | 2.30 | OS 0% + GS 90% |

The properties change linearly and depend only on the amount of glass sand (GS). The pictures show fragments of bricks after the compressive strength test. The photographs (Figures 7 and 8) show the color, texture, and porosity for the reference sample (Figure 7) and for the sample modified by 90% glass sand (GS). The macroscopic analysis (for samples with 90% OS and samples with 90% GS) shows that the reference sample (Figure 7) is characterized by greater brittleness and porosity compared to a sample made entirely of glass sand (90% GS). The brick modified by 90% glass sand (GS) is harder, uniform, and has a more even external surface. Macroscopic tests were confirmed by mechanical tests using a hydraulic press and analysis of these tests (Figures 9 and 10).

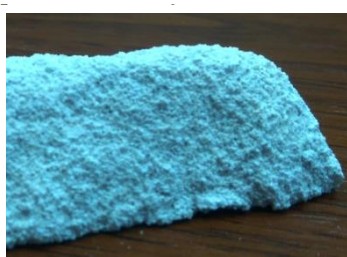

**Figure 7.** Brick (fracture) with quartz sand.

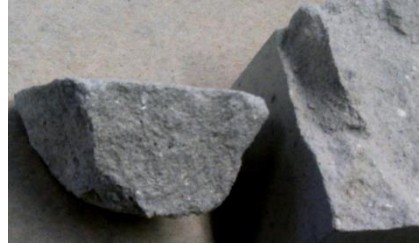

**Figure 8.** Brick (fracture) with glass sand (GS).

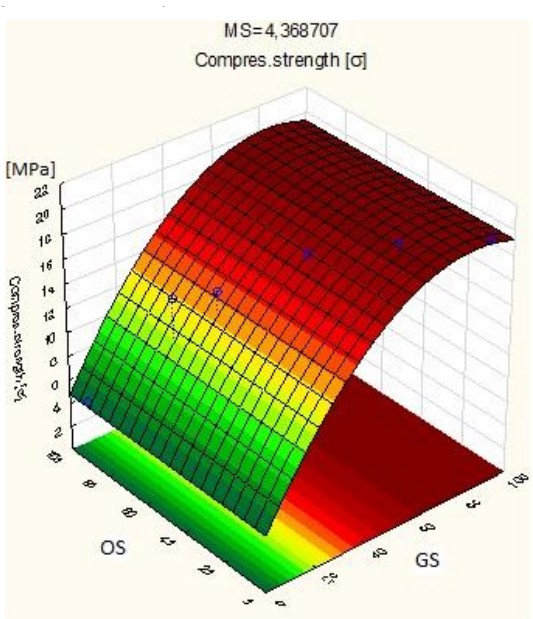

**Figure 9.** Graph of the regression function and its projection with marked experiment plan points of compressive strength value of silicate products (glass sand (GS) content from 0% to 90%).

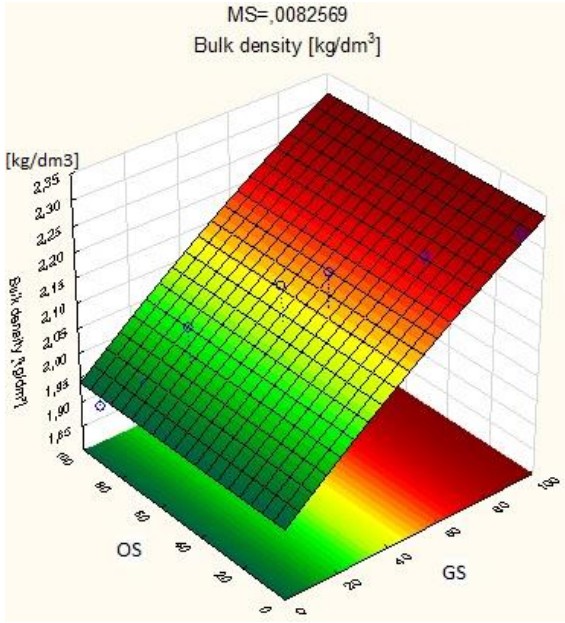

**Figure 10.** Graph of the regression function and its projection with marked experiment plan points of bulk density value of silicate products (glass sand (GS) content from 0 to 90%).

The most advantageous modification is the introduction of glass sand into the mass. A brick with high external resistance, high density, and no diminished strength is created (also considering the possibility of making a mistake during the production process). The impregnability of the modified bricks was reduced by 0.5%–1.0% compared to the reference sample (which is due to the greater humidity of the glass sand. Therefore, due to the low degree of differentiation in this property, mainly the strength and density were taken into account).

For compressive strength test as part of the adopted model, there is an estimation error equals to MS = 4.36707, which means that each of the results may be subject to an error of above 2 Mpa ($MS^{1/2}$ = 2.089 Mpa). This may be due to the shorter autoclaving time under laboratory conditions (5 h) compared to traditional silicate bricks production (8 h). The relationship during the tests varies linearly and depends only on the amount of GS in the silicate mass. The estimation error depends on the shape of the model. This may be due to the shorter autoclaving time under laboratory conditions (5 h) compared to traditional silicate bricks production (8 h). The bulk density test showed no differences between laboratory and industrial production.

## 3.2. Structural and Microstructural Analysis

After the mechanical and physical analysis for the research of the structure and microstructure, the following samples were used: the reference sample on the basis of crystalline sand (Figure 7) and samples on the basis of glass sand (GS). The XRF analysis of the composition of lime, reference sample, and sand glass made it possible to determine the quality of applied modifiers and their impact on the lime–sand mass (Table 3).

**Table 3.** Analysis of the composition of lime and glass compounds.

| XRF-CaO | | | XRF-Reference | | | XRF-GS | | |
|---|---|---|---|---|---|---|---|---|
| **Final Weight CaO:** | **7.7113 g** | | **Final Weight SiO$_2$:** | **7.7073 g** | | **Final Weight GS:** | **7.7046 g** | |
| LOI (%): | 0.018 | | LOI (%): | 0.018 | | LOI (%): | 1.304 | |
| Compound | Value | Unit | Compound | Value | Unit | Compound | Value | Unit |
| SiO$_2$ | 1.691 | % | SiO$_2$ | 100.306 | % | SiO$_2$ | 71.2 | % |
| TiO$_2$ | 0.026 | % | TiO$_2$ | 0.022 | % | TiO$_2$ | 0.1 | % |
| Al$_2$O$_3$ | 0.342 | % | Al$_2$O$_3$ | 0.091 | % | Al$_2$O$_3$ | 1.8 | % |
| Fe$_2$O$_3$ | 0.182 | % | Fe$_2$O$_3$ | 0.061 | % | Fe$_2$O$_3$ | 0.4 | % |
| Mn$_3$O$_4$ | 0.024 | % | Mn$_3$O$_4$ | 0.003 | % | Mn$_3$O$_4$ | 0 | % |
| MgO | 0.921 | % | MgO | 0.018 | % | MgO | 1.1 | % |
| CaO | 96.034 | % | CaO | 0.084 | % | CaO | 10.6 | % |
| | | | K$_2$O | 0.049 | % | Na$_2$O | 12.4 | % |
| | | | P$_2$O$_5$ | 0.006 | % | K$_2$O | 0.6 | % |

### 3.2.1. Structure Analysis

Structure and microstructure investigations of traditional bricks (reference) and bricks modified by glass sand (90 GS) after the autoclaving process was performed. The results of the phase composition with the qualitative analysis of the reference and 90 GS samples are presented in Figures 11 and 12. In the diffractograms of references samples and 90 GS, there is an increased background in the range of 2θ: 10–40° (called amorphous halo) and reflections which indicates the presence of crystalline phases in these samples (Figures 11 and 12).

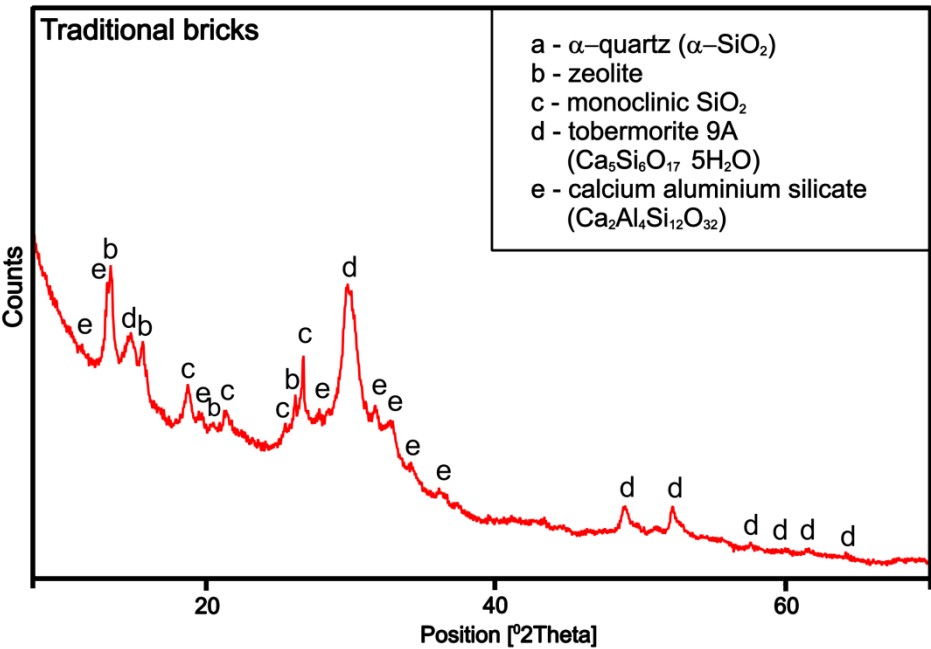

**Figure 11.** XRD analysis of the traditional bricks.

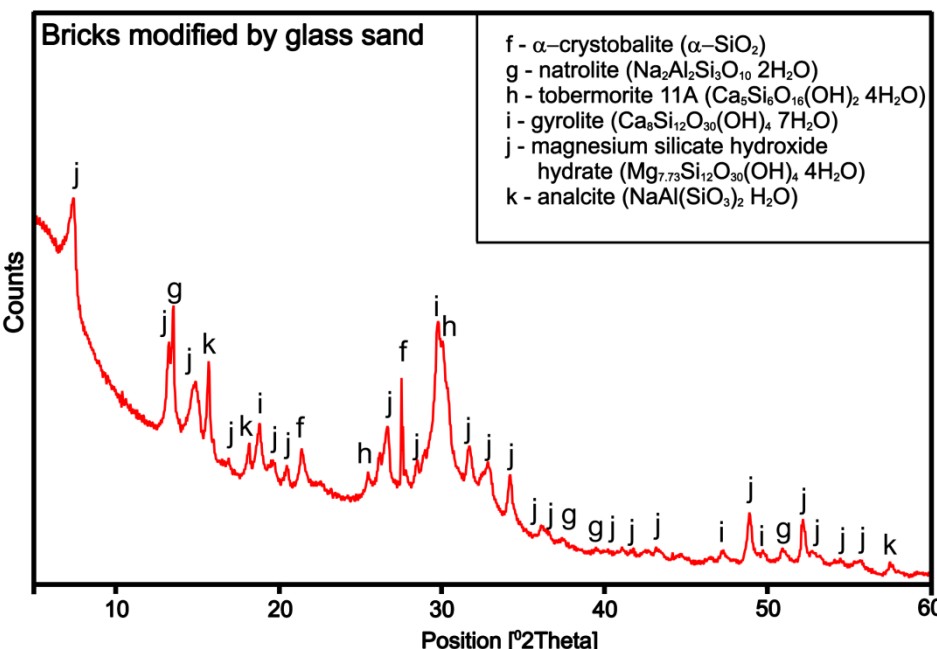

**Figure 12.** XRD analysis of the bricks modified by glass sand (90% of GS).

Figures 11 and 12 show the range of the 2-theta angle for which peaks were found in the diffraction pattern of measurement samples. Below 5 2 there were no reflections recorded on the diffraction pattern of the traditional brick sample, therefore the diffraction pattern in Figure 11 was presented in the 10–70 2 range. In the same way, could explain the range of 2 in Figure 12. Above 60 2 there were no reflections recorded on the diffraction pattern of the brick modified by sand glass sample.

The qualitative identification of the phase composition of the samples was performed with reference to the ICDD PDF-2 database.

Different range was used because no significant peaks were seen in the reference sample. Tobermorite was particularly considered because it is characteristic of autoclaved products.

In the reference sample together with the amorphous phase following crystalline phases have been occurring: $\alpha$-quartz ($\alpha$-SiO$_2$, PDF 01-080-2147), zeolite (SiO$_2$, PDF 01-073-3412), monoclinic SiO$_2$ (PDF 01-082-1563) tobermorite 9A (Ca$_5$Si$_6$O$_{16}$(OH)$_2$, PDF 04-012-1761), and calcium aluminum silicate (Ca$_2$Al$_4$Si$_{12}$O$_{32}$, PDF 04-017-9612)–Figure 11. In publications [42,43], the conditions of correct gyrolite synthesis were discussed, which was reflected in our studies (sample 90 GS). The structure of synthetic gyrolite was similar to the crystal structure natural gyrolite and the calculated crystallite size of gyrolite varied in the range of 10–50 nm, depending on synthesis conditions.

### 3.2.2. Microstructure SEM Analysis

The interpretation of the microstructure was applied with the use of a scanning electron microscope (SEM) with EDS spectrum because this type of bricks is a product of the hydration process. Calcium silicates hydrated with different degrees of structure occur in autoclaved bricks (which depends on the temperature and pressure in the autoclave). The reference samples produced on the basis of quartz sand (Figures 13–16) and products modified on the basis of glass sand (90% GS, Figures 17–22) were subjected to analysis. All the samples were sprayed with carbon during the preparation for this test. The system C–S–H (amorphic phases) is created as the result of the reaction of silica with water. Tobermorite is hydrated silica with an ordered structure. Amorphic phases show larger specific surfaces than crystalline phases. The more ordered the structure is, the smaller the specific surface. In autoclaved products modified by glass sand, where sodium (Na from recycled glass) is present, phases different than C–S–H or tobermorite were expected (because of high temperature and high pressure). Tobermorite and C-S-H phase are called differently. The phase is the C-S-H phase can may crystallize towards C-S-H I (tobermorite) or C-S-H II (jennite) phase [47,48]. Because of the high temperature and the creation of a new bond, we may observe the presence of another phase—probably natrolite, gyrolite (Figure 18), or analcite. Natrolite is formed at 373 K–473 K (99.85–100.85 °C) and with sodium substitution. In natural minerals, gyrolite may, in turn, be formed with calcium or sodium substitution.

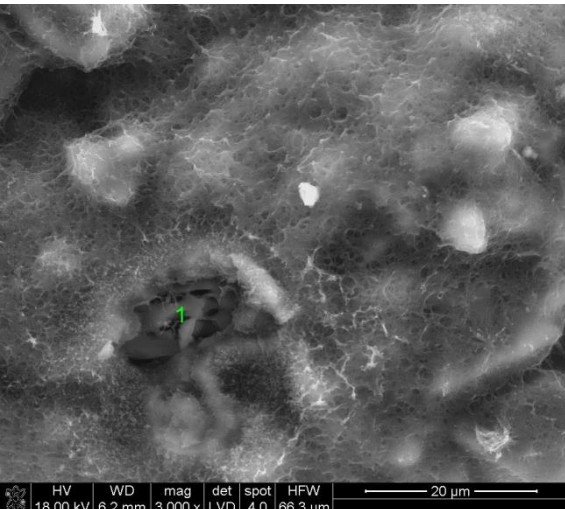

**Figure 13.** Reference sample based on quartz crystalline Sand (1-SiO$_2$, 2-C–S–H).

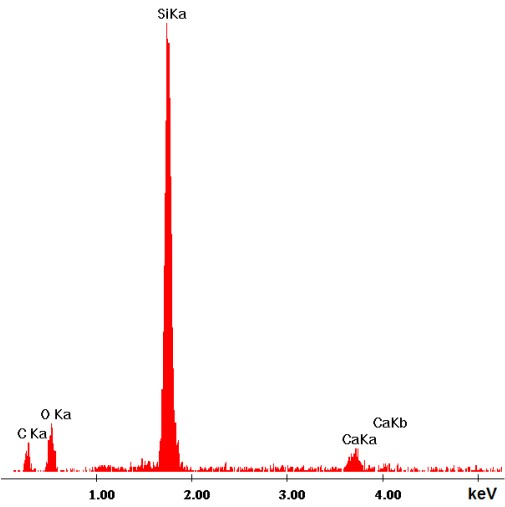

**Figure 14.** EDS spectrum in item p1 of sample based on quartz crystalline sand.

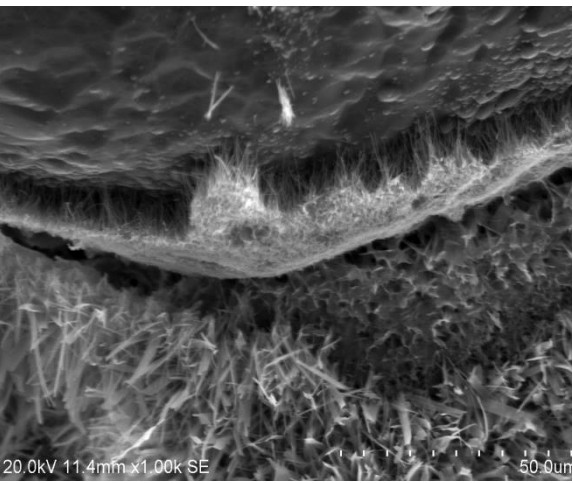

**Figure 15.** Reference sample based on quartz crystalline sand.

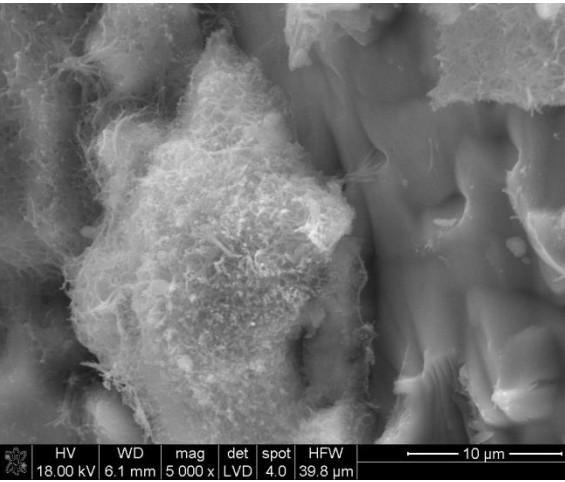

**Figure 16.** Reference sample based on quartz crystalline sand.

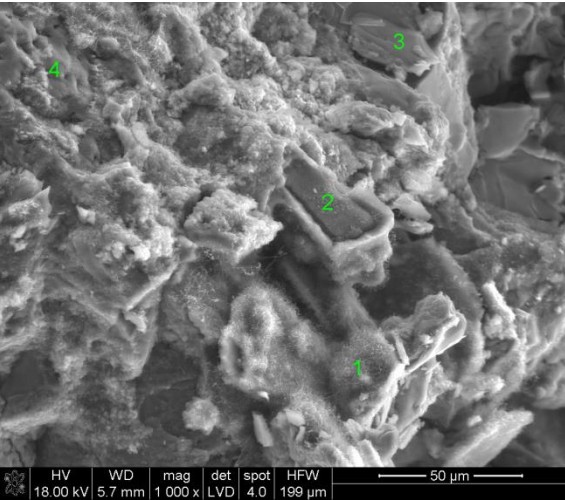

**Figure 17.** SEM image of the sample with glass sand (90% GS).

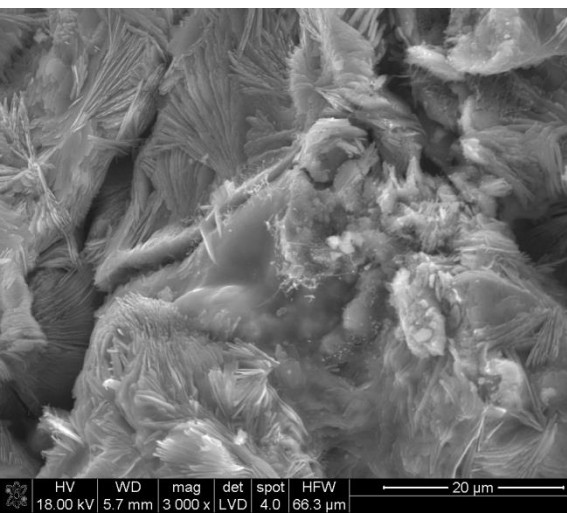

**Figure 18.** SEM image of the sample with glass sand (90% GS, Gyrolite).

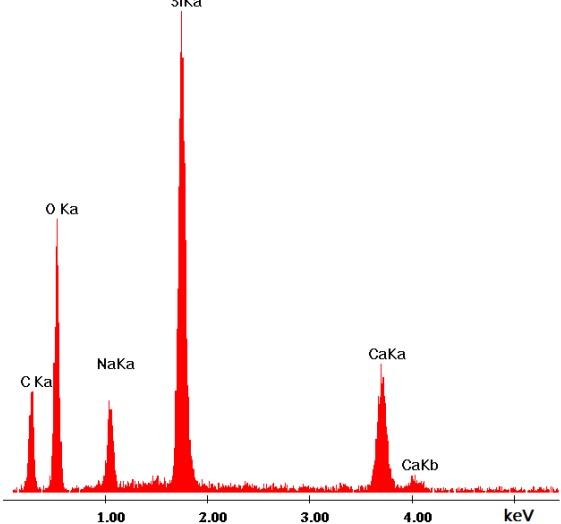

**Figure 19.** EDS spectrum in an item p.1 of the sample with glass sand (90% GS).

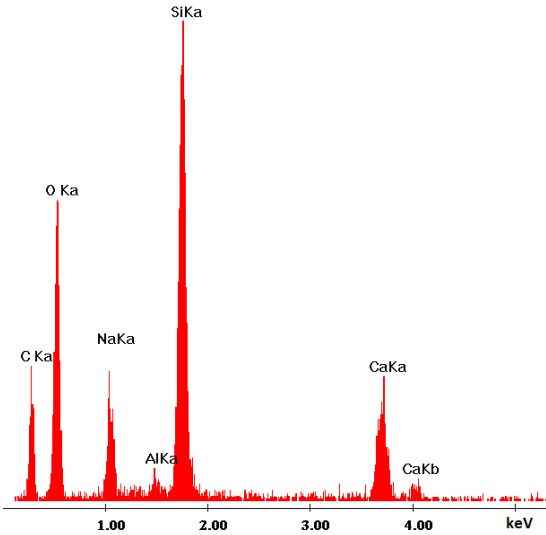

**Figure 20.** EDS spectrum in an item p.2 of the sample with glass sand (90% GS).

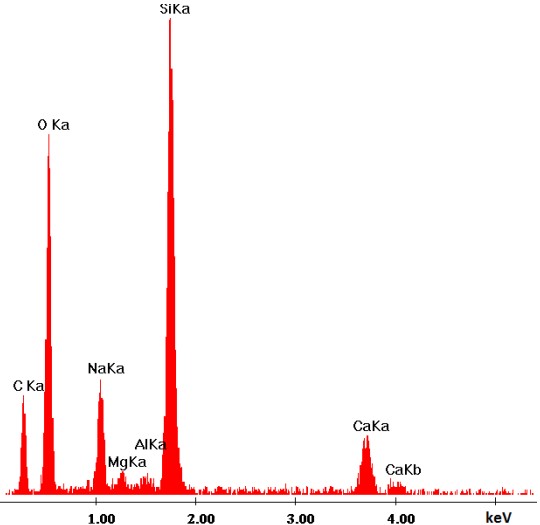

**Figure 21.** EDS spectrum in an item p.3 of the sample with glass sand (90% GS).

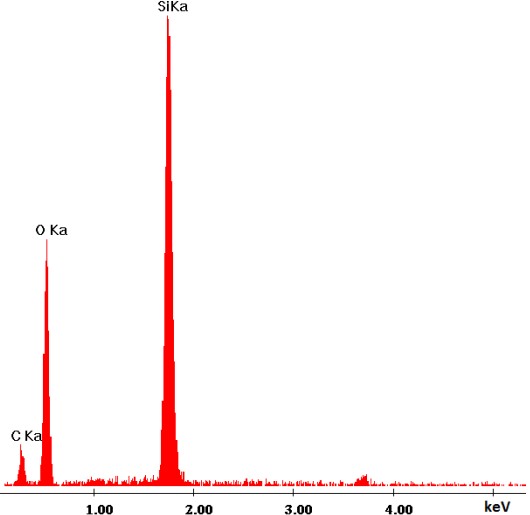

**Figure 22.** EDS spectrum in an items p.4 of the sample with glass sand (90% GS).

The elemental analysis in the form of the EDS spectrum for a reference sample at a given point is presented below.

In the microstructure based on the EDS spectrum, changes in the quality and quantity of hydrated calcium silicates can be observed. The presence of sodium facilitates the synthesis of gyrolite in bricks made on the basis of glass sand. This type of brick in macroscopic and strength analysis has a higher hardness compared to traditional bricks based on quartz sand. It happens, however, that a glass brick, during the autoclaving process, is emphasized by about 1 mm on one surface (above the form). It probably has a connection with too small graining (no fraction in the range of 1–2 mm) and the presence of additional elements (Sodium).

Microscopic observations of EDS showed the participation of significantly different mineral fractions with uneven particle sizes. However, the microstructure was more consistent and had less porosity. The autoclaving process was reduced by 1 h (1 h heating + 5 h autoclaving +1 h cooling), which could have contributed to the lack of proper development of the next phase (Figure 18). Sodium is visible at all points marked on EDS photographs. Aluminum (Al) comes from sand (industrial deposits).

EDS analysis can be qualitative and quantitative. Quantitative EDS mapping is possible for flat and polished samples. Tests were carried out for porous sputtered samples. The x axis is [keV], and y axis is intensity (EDS during testing with a scanning electron microscope-according to the methodology).

## 4. Conclusions

This paper aims to show the basic differences between the use of crystalline quartz sand and glass sand applied to brick modification and which has an amorphous structure (similarly to fly ash that has a beneficial effect on concrete durability and structure).

The STATISTICA program was used to determine the mechanical and physical properties of the sustainable bricks, especially compressive strength [MPa] and bulk density [kg/dm$^3$]. The tests showed the improvement in the strength characteristics of sand–lime bricks (mortar), mainly resulting from the presence of glass compounds, in particular glass sand (up to 1 mm particle size). In this modification, under laboratory conditions, with an autoclaving time of 5 h (1 + 5 h + 1), the compressive strength of the reference material was 5.25 MPa and the density was 1.92 kg/dm$^3$ (sufficient for the construction material). However, for the same conditions, the compressive strength of material modified by glass sand (GS) from recycled glass was 20.23 MPa (industrial scale), and density was 2.30 kg/dm$^3$. Impregnability and water absorption remained at the similar level. The results also show that the more glass sand GS is added, the better resistance and density will be achieved. The only disadvantage of this modification is the impact of water and vapor on recycled glass (silica) under hydrothermal conditions. Glass compounds are amorphous substrates which give different results in the absence of precision during brick production. Additional studies are needed concerning the effect of the pressure of water vapor and the temperature inside the autoclave on the stability and the physical properties of sustainable material with glass compounds. In this case, it seems necessary to check the C/S ratio (Si/Ca), and check precisely which phase occurs in the sustainable brick and how to correlate all phases that form with the physical and mechanical properties of the bricks. Both materials (reference and glass sand-containing material) vary in appearance and porosity—glass sand samples are more uniform on the external surface and are characterized by lower porosity than the samples with quartz sand. Lime hydration temperature in the presence of sand is also different; for quartz sand it is about 80 °C, but for amorphous glass sand it is only about above 42 °C (depending on the amount of GS). The sodium present in GS (glass sand) may cause the material to swell (Figure 4). It may depend on the composition of the glass compounds. In this case, additional chemical analysis and test will be necessary. Modification by glass sand (GS) also leads to crystallization of other phases such as sodium aluminosilicate hydrated (basic sodium calcium silicate hydrate: N–C–S–H), natrolite ($Na_2Al_2Si_3O_{10}2H_2O$), tobermorite 11A, and gyrolite ($NaAl(SiO_3)2H_2O$). The phase that occurs in the sustainable brick has to be identified. The next task, currently in preparation, is the investigation of pressure and temperature related thermodynamic changes that affect the calcium silicate hydrate.

Thermodynamic modeling is primarily applied to concretes up to 100 °C. The authors of this paper intend to apply it to autoclaved materials. This problem needs further research, thus the analysis will be continued.

**Author Contributions:** Conceptualization, A.S.; Methodology, A.S., M.S.; Software, A.S., M.L.; Validation, A.S., M.S.; Formal Analysis, A.S.; Investigation, A.S.; Resources, A.S., M.S.; Data Curation, A.S., M.L.; Writing—Original Draft Preparation, A.S.; Writing—Review & Editing, A.S., M.L., M.S.; Visualization, A.S. and M.S.; Supervision, M.S.; Project Administration, A.S.; Funding Acquisition, A.S, M.L.

**Funding:** The article was created as part of the implementation of projects: the MINIATURA 2 Grant (funded by National Science Center, Cracow, Poland: Register no.: 2018/02/X/ST8/00544, application ID: 409666) and Project No. 2015/19/N/ST8/00486 (funded by National Science Centre).

**Acknowledgments:** First author would like to acknowledge to: M.Balonis and G.N.Sant (University of California Los Angeles, USA), Tagnit-Hamou A. (University of Sherbrooke, Canada), Z.Koruba, Z.Piotrowski, A.Szmidt, R.Dachowski (Kielce University of Technology), T.Stepien and L.Kotulski, W.Robak (Silikaty Ludynia) for scientific cooperation and support.

**Conflicts of Interest:** The authors declare no conflict of interest.

### Nomenclatures

| | |
|---|---|
| OS | Quartz sand |
| GS | Glass sand |
| XRF | X-ray Fluorescence |
| SEM | Scanning Electron Microscope |
| XRD | X-ray Powder Ddiffraction [keV] |
| $\varrho_o$ | bulk density [kg/dm$^3$] |
| $\delta$ | compressive strength [MPa] |
| $m_s$ | the mass of the dry sample |
| $n_w$ | impregnability of weight |
| $m_n$ | the weight of the wet samples |
| m | the weight of the dry samples |

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
