# Peer review of "A Sustainable Autoclaved Material Made of Glass Sand"

_buildings, doi:10.3390/buildings9110232_

Round 1

Reviewer 1 Report

The present paper reports the improvement of the building materials by addition of glass sand during the hydrothermal synthesis. The authors analyzed the microstructure of resulting products by XRD, SEM and XRF analyses. It is entirely appropriate for Buildings. I basically agree with significance of the paper. The following two points should be considered prior to publication.

With regards to the XRD results, the intensity of magnesium silicate hydroxide was rather intense although the magnesium content was low. I feel that the XRD result was not consistent with the XRF result. More detail explanation should be added.

With regard to the microstructure characterization, the characterization result was insufficient. Information on domain size of each component and EDS mapping should be added. Discussion is also insufficient to understand the improved property of present building materials.

Author Response

Dear Reviewer,

Thank you so much for the review and your support.

Best Regards,

Authors

Reviewer 2 Report

Line 27: “and offer many advantages”, the authors need to list the advantages Line 36-37: needs references Line 51-53: needs reference Line 59: add a space between unit and figure (56g) Line 58: remove “here” Rather than denoting “reaction 1” please use (equation 1). Similar for the other reactions The literature could be narrowed down and be straight to the point. Also the aims of the research is missing at the very last paragraph of the introduction Line 103: “autoclaving temperature was equal to 199-200 °C” is a repeating sentence Line 116: “2 theta scanning range” what was the exploration range? Needs to be added. The method section can be divided into subheadings. The authors wrote all analytical techniques under a heading. Which data base was used for XRD analysis in Highscore plus? Line 119-120“Concentration of a particular element was determined by measuring its lines intensity”. What was the ground to choose the intensity as an indicator for concentration? Line 123:” X-ray diffraction (XRD)” this was already defined Line 126: “5 x 5 x 5x cm” revise this Please use either “mm” or “cm”. Please check throughout the manuscript. Line 140-143: Have the authors tested the particle size using Mastersizer or similar techniques/machine? If yes, the graph must be added. Please combine Figs. 1 and 2. Figure 6: the caption says: “Autoclaved materials before loading into the autoclave”. Is the product already autoclaved? If so, why you re-autoclaved it? Material section usually comes as the first heading in material and method section Line 192: “0,5-2 mm 40-50%)” check the punctuation please Line 246: “The pictures show bricks and their cracks”. There is no crack in the pictures. Formula 6: why you have used this regression equation? Also why did you not consider the synergic effect on the regression model? Line 238 and 243: Table 1, not “Tab.1” STD or SE must be included in Table 1. The first two columns can be removed in table 1 as they provide the same information as the last column. Fig 9 has very low quality and Z axis is not readable. Line 261: check the caption (there is a dot which should be removed) Combine Fig. 9 and 10. Also these two figures give the same information as Table 1 in Line 252. What is the pint of having them as you did not consider the interaction effect (GS*OS)? -There are two table 1 in the manuscript. Check the cross referencing as well. 11 and 12 must be combined. Line 292: SEM was already defined 13 and 14 must be combined 14 : y axis is missing, axis titles are missing Base on Fig. 13, the authors did EDS for point 1 which is located in a hole. EDS is not working properly on such points and areas as the detector cannot collect all diffracted energy so the results might not be reliable. 15 and 16 must be combined 17, 19,20,21, 22 must be combined. Since there is no unexpected material in the fabricated brick, what was the reason to do EDS analysis as all compounds were expected to be there based on the mix design? EDS analysis does not add value to your research. The paper lacks a discussion. The authors just presented the experimental data without critical discussion on why such results have been obtained. There is no comparison study with the literature to see whether the research has improved the concept of autoclaved brick or not.

Author Response

Dear Reviewer,

Thank you very much for the review and comments. We’re sending answers below.

With regards to the XRD results, the intensity of magnesium silicate hydroxide was rather intense although the magnesium content was low. I feel that the XRD result was not consistent with the XRF.

The XRF result (MgO 0.018%) and the XRD result were performed correctly.

The XRF test was carried out 7 days after the production of autoclaved bricks (2 series: A-reference, 6 samples and B-modified by glass sand (90%GS), 6 samples as well).

The first XRD test was made in Sherbrooke (University of Sherbrooke, Canada) 14 days after the date of brick production.

The results were not very accurate due to the database and due to the use of an amorphous additive, therefore, for this paper the test was reproduced on AGH University in Cracow (Poland) using a more accurate database, taking into account hydrated calcium silicates formed in hydrothermal conditions.

We have samples A and B series all the time and files from each test.

It has also been proven that under the influence of time and temperature, thermodynamic changes occur in a sample modified with glass sand (amorphous phases crystallize). However, this comparison will be the subject of another paper in this series of studies.

In addition, the sand for modification came from a sand mine at the Brick Production Plant in Ludynia (Poland), hence the presence of MgO.

XRD is also a qualitative study of a small area of the tested sample.

The main aspect of research is the possibility of using recycled glass sand in building materials created in hydrothermal conditions.

At the initial stage of the production of autoclaved bricks, a lime slaking process takes place, where the temperature rises up to 90oC (which depends, among others, on the quality of lime CaO, sand moisture and water quantity) and which affects the later durability of the material.

In the case of the use of glass sand, the situation is different despite a similar manufacturing proces:

First of all - we have an amorphous structure substrate, which is theoretically a more reactive additive; Secondly, the reaction temperature of water and lime in the presence of glass sand (90% GS) drops by half. It has been proven that the 'new product' after production gets the strength required for this type of brick. Now this product should be observed over time by performing the same tests continuously and cyclically, and any changes should be recorded.

The next part of the research is currently being prepared, concerning especially thermodynamic changes. Because the C-S-H phase is especially present in concretes material and is thermodynamically stable at temperatures up to 30oC.

In the case of autoclaved bricks, this phase is the minimum for tobermorite (reference sample) and for natrolite or/and gyrolite (bricks with Glass Sand, due to the presence of sodium).

Therefore, an interesting aspect may be also the formation of the M-S-H phase with admiration of MgO, instead of the C-S-H phase.

If the Reviewer allows, we will describe it in the next paper.

With regard to the microstructure characterization, the characterization result was insufficient. Information on domain size of each component and EDS mapping should be added. Discussion is also insufficient to understand the improved property of present building materials.

EDS analysis can be qualitative and quantitative.

Quantitative EDS mapping is possible for flat and polished samples. The tests were carried out, however, for porous samples with sputtering.

However, quantitative tests are carried out for both reference polished samples as well as 50/50% and 90% GS, but they are also part of the research concerning only the microstructure of a ‘new glass sand-modified brick’.

This paper aims to determine the basic properties that occurred (or lack of them) during the replacement of crystalline quartz sand (90% QS) by recycled glass sand (90% GS), which is additionally characterized by an amorphous structure.

Thank you so much for your comments and support.

Best Regards,

Authors

Round 2

Reviewer 1 Report

I understand the reviewer's comment.

Author Response

Dear Reviewer,

Thank you so much for the review and your support.

Response 1:  Is the research design appropriate?

Materials created during hydrothermal treatment behave differently than concrete which gains compressive strength after 28 days. The structure of these materials is also different.

The type of hydrated calcium silicates depends on the amount of binder (cement, lime, binder/gypsum). In autoclaved bricks normally we use 7% of lime and 90% of quartz sand, so the C / S molar ratio will be less than 0.83. Adding to this process temperature (around 200oC) and pressure, the phase that forums is tobermorite.  If there was more lime, it would be possible to analyze jennite. The C-S-H phase crystallizes under the influence of temperature, and that’s the reason why we are focused on tobermorite.

Response 2: Are the methods adequately described?

The methods have been selected to show the first stage of research for the use of 90% glass sand in autoclaved bricks.

This is the first manuscript in the series of articles (about glass sand in autoclaved materials) that we would like to present.

This paper aims to show the basic dependence between the use of crystalline quartz sand, and then replace this type of sand by glass sand, which has an amorphous structure (for example: fly ash with an amorphous structure has a beneficial effect on the durability and structure of concrete due to the presence of the C-S-H phase. However, in hydrothermal treated materials the C-S-H phase is the deficit phase. The basic phase is tobermorite).

Additional microscopic tests, reaction thermodynamics and stability will be a continuation of the tests presented here.

Dear Reviewer,

The drawings were reorganized because this was the suggestion of Reviewer No. 3.

Best Regards,

Authors

Reviewer 2 Report

The authors randomly addressed the comments. I recommend the authors carefully read the previous round comments and reply to the comments/suggestions individually.

Author Response

Dear Reviewer,

Thank you so much for the review and your support.

Response 1:  Does the introduction provide sufficient background and include all relevant references?

The production process of this kind of brick in is similar to autoclaved cellular concrete (AAC), but differs from traditionally produced concrete because it is produced under hydrothermal conditions (200oC). In AAC’s, crystalline phases such as tobermorite are formed contrary to nearly amorphous gel calcium silicate hydrate (known as gel C-S-H, and next C-S-H phase) which is found in traditional concrete. Because of these points, the study describes the dependence between the basic physio-mechanical and chemical properties of sand-lime materials which undergone hydrothermal treatment (autoclaving process) and which were modified through the introduction of glass components (glass sand “GS”).

The use of glass components in concrete is already wide and known in the world (but the use of glass components in autoclaved bricks is less popular, that’s why we would like to explain this process).

References relate mainly to similar research conducted on autoclaved materials (2,5,7,8,13-20,22). The other references relate mainly to the results for research and materials where Glass Sand was used and gyrolite or natrolite were formed.

Response 2:  Is the research design appropriate?

Materials created during hydrothermal treatment behave differently than concrete which gains compressive strength after 28 days. The structure of these materials is also different.

The type of hydrated calcium silicates depends on the amount of binder (cement, lime, binder/gypsum). In autoclaved bricks normally we use 7% of lime and 90% of quartz sand, so the C / S molar ratio will be less than 0.83. Adding to this process temperature (around 200oC) and pressure, the phase that forums is tobermorite.  If there was more lime, it would be possible to analyze jennite. The C-S-H phase crystallizes under the influence of temperature, and that’s the reason why we are focused on tobermorite.

Response 3: Are the methods adequately described?

The methods have been selected to show the first stage of research for the use of 90% glass sand in autoclaved bricks.

This is the first manuscript in the series of articles (about glass sand in autoclaved materials) that we would like to present.

This paper aims to show the basic dependence between the use of crystalline quartz sand, and then replace this type of sand by glass sand, which has an amorphous structure (for example: fly ash with an amorphous structure has a beneficial effect on the durability and structure of concrete due to the presence of the C-S-H phase. However, in hydrothermal treated materials the C-S-H phase is the deficit phase. The basic phase is tobermorite).

Additional microscopic tests, reaction thermodynamics and stability will be a continuation of the tests presented here.

Response 4: Are the results clearly presented?

In this paper first presents the production process of traditional sand-lime brick, the basic reactions that occur during the production process of this type of brick, and the autoclaving process (“During the next 6-12 hours of autoclaving, the lime reacts chemically with sand and the mixture undergoes the process of recrystallization (usually it takes 8 hours, 1hour heating, 8 hours autoclaving and 1-hour cooling”).

Next, point 2 describes the materials, research method, the basicity of the tests and the autoclaving process for this particular test (autoclaving lasted 5 hours due to the possibilities of the laboratory autoclave).

The photographs Fig.3 and Fig.4 show bricks manufactured using 90% OS (quartz sand) and next to bricks using amorphous glass sand (90% GS).

First we showed physical changes, then mechanical properties (compressive strength) as so-called external features, and then we presented the microstructure of these materials.

Microstructure tests are particularly important, That’s why we would like them to be a continuation of our further research.

We would like to keep this research order and results that are correct.

Response 5: Are the conclusions supported by the results?

Research goal =>

The main aspect of the research is the possibility of using recycled glass sand in building materials created in hydrothermal conditions.

At the initial stage of the production of autoclaved bricks, a lime quenching process takes place, where the temperature rises up to 90oC (which depends, among others, on the quality of lime) and which affects the later durability of the material.

When the replacement of crystalline sand with glass sand occurs, the following changes occur:

First of all - we have an amorphous structure substrate, which is theoretically a more reactive additive; Secondly, the reaction temperature of water and lime in the presence of glass sand drops by half ( around 39-46 (depending on the amount of GS - the low temperature is during mass modification by 90% GS)).

It has been proven that the 'new product' after production gets the strength required for this type of brick.

The product should now be observed over time by performing the same tests continuously and cyclically, and any changes should be recorded.

The next part of the research is currently being prepared. The next paper is focused on thermodynamic properties, because the C-S-H phase is present in concretes material and is thermodynamically stable at temperatures up to 30oC. But in sand-lime bricks the dominant phase is tobermorite and other crystalline phases. This difference is important for the research on microstructure of the bricks.

Thuse, the next task, currently in preparation, is the investigation of pressure and temperature related thermodynamic changes that affect the calcium silicate hydrate. Thermodynamic modelling is primarily applied to concretes up to 100oC. The authors of this paper intend to apply it to autoclaved material. This problem that needs further research, thus the analysis will be continued.

Dear Reviewer,

The drawings were reorganized because this was the suggestion of Reviewer No. 3.

Thank You and Best Regards,

Authors

Round 3

Reviewer 2 Report

The authors addressed the default questions in the review panel (e.g. Does the introduction provide sufficient background and include all relevant references?; Is the research design appropriate?; Are the methods adequately described?; Are the results clearly presented?; Are the conclusions supported by the results? etc) instead of addressing the reviewers’ comments. I strongly recommend the authors carefully address all points raised below which most of them were available in the previous two rounds too.  

Line 12-130: XRD exploration range (2theta) must be included. Why two different XRD ranges were tested (10-70 degrees in Fig 11 and 5-60 degrees in fig 12)? Since there is a compounds found after 60 degrees (Ca5Si65H2O), how do you justify there is no such a compound in brick modified by glass sand? Question from previous two rounds: “Which data base was used for XRD analysis in Highscore plus?” is should be added Line 138: 50 x 50 x 50 mm not 50 x 50 x 50 xmm Comment from previous two rounds: Figure 6: the caption says: “Autoclaved materials before loading into the autoclave”. Is the product already autoclaved? If so, why you re-autoclaved it? Question from previous two rounds: Equation 6: why you have used this regression equation? Also why did you not consider the synergistic effect on the regression model (GS*OS)? Standard deviation or standard error must be included in Table 1 for compressive strength column 9 and 10 should be combined. Also the x,y,z axes in these two figures are not readable. Comment from previous two rounds “Figure 14: y axis is missing” Y axis is missing in all EDS graphs. Question from previous two rounds: Base on Fig. 13-14, the authors did EDS for point 1 which is located in a hole. EDS is not working properly on such points and areas as the detector cannot collect all diffracted energy so the results might not be reliable. There are too many figures (22 figures) in this paper which most of them can be combined. I strongly suggest the authors combined the following figures and lable them accordingly. For example Fig1 and 2 can be combined; Fig 3 and 4; Fig 7 and 8; Fig 9 and 10; Fig 11 and 12; Fig. 13 and 14; Fig 15 and 16; Fig 17 and 18; Fig 19 and 22. Comment from previous two rounds :The paper lacks a discussion. The authors just presented the experimental data without critical discussion on why such results have been obtained. There is no comparison study with the literature to see whether the research has improved the concept of autoclaved brick or not.

Author Response

Line 12-130: XRD exploration range (2theta) must be included.

The phase analysis of the traditional brick and brick modified by sand glass samples have been measurement in the 5-70° range of 2q.

Why two different XRD ranges were tested (10-70 degrees in Fig 11 and 5-60 degrees in fig 12)? Since there is a compounds found after 60 degrees (Ca5Si65H2O), how do you justify there is no such a compound in brick modified by glass sand?

Figs. 11 and 12 show the range of the 2 theta angle for which peaks were found in the diffraction pattern of measurement samples. Below 5° 2q there were no reflections recorded on the diffraction pattern of the traditional brick sample, therefore the diffraction pattern in Figure 11 was presented in the 10-70° 2q range. In the same way, could explain the range of ° 2q in Fig. 12. Above 60° 2q there were no reflections recorded on the diffraction pattern of the brick modified by sand glass sample.

Question from previous two rounds: “Which data base was used for XRD analysis in Highscore plus?”

Qualitative identification of the phase composition of the samples was performed with reference to the ICDD PDF-2 database.

Line 138: Corrected. Figure 6: We answered on this question after first review.

Figure 6 shows the molds with a fresh mixture (sand+lime+water) after lime hydration process and compression using a press, but before autoclaving process (accorging to Figure 5).

As figure 5 shows, the bricks are autoclaved ONLY ONE TIME. Fresh sand-lime mass AFTER THE LIME HYDRATION PROCESS, is placed in molds and then placed in an autoclave for 5 hours (laboratory conditions) or for 8 hours (industrial conditions).

Equation 6

Ï­ = A0 + A1 (OS) + A2 (OS)2 + A3 (GS) + A4 (GS)2         

Orthogonal compositional plan type 3k (with k = 2), i.e. full two-factor experiment was applied in order to carry out the experiments both in the compression strength test and bulk density test. For each factor correlation, six parallel tests have been conducted. The methodology of experiment and obtained results are shown in Table 2.

No standard error was used because the lime hydration temperature, strength and density change linearly and depend only on the amount of GS by weight (0-90% GS by weight).

The Statistica program was used to better visualize changes in physical and mechanical properties.

For Figures 9 and 10, quality is related to sending data from Statistica to Word.

Table 1 -should be combined:

I'm sorry but Table 1. shows composition of the traditional sample and sample (first column) with glass sand (second column). It can not be combined.

Table 2. Plan of the experiment. Bricks with GS & traditional quartz sand (OS).

Also, we cannot connect the columns because they represent the amount of quartz sand and at the same time glass sand in lime-sand mass (from 0 to 90%).

Quartz sand is always the basis of reference products. Glass sand appears as an amorphous modifier and the properties of the 'new brick' depend on it.

Tables 1 and 2 do not contain columns 9 and 10.

Figure 13-14 : We answered this question in the first review:

The image shows sand that is surrounded by a C-S-H phase. This phase then crystallizes into tobermorite (otherwise C-S-H I).

Therefore, in Figure 13 sand is visible and this is not a 'hole', but SiO2 is surrounded by a phase.

The figures 1-22 were combined as suggested after the first review. Some figures have not been combined to avoid loss of quality.

The discussion has been completed. We add that glass sand and glass powder is used on a wide scale in concretes, but autoclaved bricks are another type of material.

One of the problems why this modification is not so common is the lack of a dominant C-S-H phase in bricks (in autoclaved products/bricks the C-S-H phase crystallizes).

In addition, sodium is present in the glass, which causes swelling (in the absence of the production process).

In this article, we present the traditional production process of autoclaved bricks. Then the laboratory method of production of autoclaved bricks modified by glass sand and analyzing the basic changes.

The X axis is [keV], and Y axis is intensity. the photos were obtained from the SEM microscope examination and we included them in the article (without corrections). The C / S ratio and the number of individual primroses in the sample is a test which is performed on polished samples. In this test we performed SEM on porous samples.

The article presents the basic results of the research. Further studies are in preparation, because under the influence of time and external conditions (temperature) thermodynamic changes took place, which are particularly visible in the microstructure of the material. However, we do not want to describe this in this article.

Discussion:

We will critically refer to the additional discussion in the next series of studies - for now we want to illustrate the changes that occur when switching from quartz sand to glass sand.

Thank you,

Authors

Round 4

Reviewer 2 Report

The authors stated “The phase analysis of the traditional brick and brick modified by sand glass samples have been measurement in the 5-70° range of 2q.” but Figure 12 shows a different range (5-60 degrees). If you found no peaks after 60 degrees, you need to show them not removing the remaining part.

The authors stated, “No standard error was used because the lime hydration temperature, strength and density change linearly and depend only on the amount of GS by weight (0-90% GS by weight).” This simply means the data is not reproducible, as authors have not used replications. In engineering/science practices (no matter brick experiment or other activities), the researchers must test at least two samples per each run to show the data has not been obtained by chance or error. So it is a common practice to use standard error and standard deviation to show the errors associated with the experiments. In case of using Design of experiment, the software will give you R2 representing how the experimental data was fitted to the numerical model. Therefore, if you used design of experiment package, you must report R2, otherwise, standard error for each run must be reported.

The authors stated: “The Statistica program was used to better visualize changes in physical and mechanical properties. For Figures 9 and 10, quality is related to sending data from Statistica to Word.” This is not acceptable. There are an array of free tools to edit the figure and rewrite the axis labels.

Sorry I meant Figure 9 and 10 must be combined not column 9 and 10.

Comment from the previous rounds “Figure 14: y axis is missing” Y axis is missing in all EDS graphs”. The authors simply ignore this comment and just replying the graphs were exported by the SEM-EDS.

Dear authors, Point 1 in Figure 13 is clearly on a lower surface than remaining part of micrograph so I called it as a “hole” or you can call it whatever you wish. That’s why I said the EDS might not be correct because all energy cannot be detected by EDS.

The authors stated that “The figures 1-22 were combined as suggested after the first review. Some figures have not been combined to avoid loss of quality.” How combining figures results in decreasing the quality?

Author Response

Dear Reviever,

I want to write that I absolutely did not ignore any of your opinions. Short time to answer meant that my answers were equally short and may be burdened with aesthetic error. I'm really sorry.

Below, I’m sending my answers to your comments.

I understand that you have access to all research techniques. Unfortunately, I did some of the research outside my university and in some cases I only have read-only files,  but not to be modified- such as photos from ststistics - I cannot modify them at this stage. I appologize for that.

That's why this article describes the basic properties, and on subsequent research I work with better scientific units, where I can have more access to research equipment and help.

in the attachment I am sending answers to your comments.

Thank You and Best Regards,

Authors

Round 5

Reviewer 2 Report

Since the authors stated that they performed most of the analyses outside the university and there is no way to modify the figures, etc., though I believe the figures' axes can be added manually using Photoshop, Corel, Paint, ...

At this stage, I support the manuscript for publication.

Please remove "SEM" from the captions of all EDS spectra because they are EDS spectra, not SEM images.